# Integrated genomics and comprehensive validation reveal drivers of genomic evolution in esophageal adenocarcinoma

Subodh Kumar[1,2,4], Leutz Buon[1,4], Srikanth Talluri[1,2,4], Marco Roncador[1], Chengcheng Liao[1,2], Jiangning Zhao[1,2], Jialan Shi[1,2,3], Chandraditya Chakraborty[1], Gabriel Gonzalez[2,3], Yu-Tzu Tai [1], Rao Prabhala[1,2,3], Mehmet K. Samur [1], Nikhil C. Munshi [1,2,3] & Masood A. Shammas [1,2✉]

Esophageal adenocarcinoma (EAC) is associated with a marked genomic instability, which underlies disease progression and development of resistance to treatment. In this study, we used an integrated genomics approach to identify a genomic instability signature. Here we show that elevated expression of this signature correlates with poor survival in EAC as well as three other cancers. Knockout and overexpression screens establish the relevance of these genes to genomic instability. Indepth evaluation of three genes (*TTK*, *TPX2* and *RAD54B*) confirms their role in genomic instability and tumor growth. Mutational signatures identified by whole genome sequencing and functional studies demonstrate that DNA damage and homologous recombination are common mechanisms of genomic instability induced by these genes. Our data suggest that the inhibitors of *TTK* and possibly other genes identified in this study have potential to inhibit/reduce growth and spontaneous as well as chemotherapy-induced genomic instability in EAC and possibly other cancers.

[1] Dana Farber Cancer Institute, Boston, MA, USA. [2] Veterans Administration Healthcare System, Boston, MA, USA. [3] Harvard Medical School, Boston, MA, USA. [4] These authors contributed equally: Subodh Kumar, Leutz Buon, Srikanth Talluri. ✉email: Masood_Shammas@DFCI.Harvard.edu

Esophageal adenocarcinoma (EAC) is the sixth biggest cause of cancer deaths throughout the world[1]. The incidence of this cancer is rising rapidly in Western countries. Moreover, the tools and strategies for early detection and treatment of this cancer are still not very effective and disease remains to have poor clinical outcome[1]. The disease is associated with a precancerous condition, Barrett's esophagus, which gradually progresses to EAC[2]. Similar to most cancers, EAC is also associated with a marked genomic instability that arises at an early stage and allows ongoing accumulation of genomic changes, some of which contribute to the development and progression of cancer[3]. Evaluation of single-nucleotide polymorphisms (SNPs) in patient genomes has shown that in addition to EAC, genomic changes can also be seen in a majority of Barrett's esophagus cases[4]. Microsatellite instability has also been detected at both the Barrett's esophagus and in EAC stages[5]. In fact changes of both the genetic and epigenetic type have been detected at Barrett's esophagus stage[6]. There is also evidence that genomic instability increases with progression of Barrett's esophagus to cancer. Consistently, the aneuploidy detected in a subset of Barrett's esophagus cases has been shown to progressively increase with progression to EAC[7]. Similarly, it has been reported that copy number changes spanning relatively large areas of genome are rare in early-stage disease but occur more frequently and involve larger areas of genome in advanced stages[8]. Moreover, it has been shown that although mutational load in precancerous Barrett's esophagus cases is lower than in EAC, it is higher than that observed in certain other cancer types[9,10].

EAC genome is highly aberrant with approximately ten single-nucleotide variations per million base pair[9]. Genomic instability and its consequences can probably be attributed to the chemoresistant nature of EAC[11]. There is also evidence that suggests genomic instability contributes to disease progression[12] and associates with poor survival in EAC[13]. Using EAC and multiple myeloma (MM) as cancer model systems, we have reported that homologous recombination (HR), a prominent DNA repair system, is spontaneously elevated in these cancers and contributes to genomic instability[14–16], development of drug resistance[16], and telomere length maintenance and tumor growth[17]. We have also shown that acid and bile, the main contents of gastroesophageal refluxate, induce HR activity in human cells[14]. Bile acids have also been shown to cause oxidative DNA damage in Barrett's esophagus cells[18]. As DNA damage can activate HR, specific microenvironment of Barrett's esophagus/EAC (exposed to acid and bile) may contribute to dysregulation of HR and genome stability.

In this study, we used an integrated genomics approach to identify mediators of genomic instability in EAC. Functional significance of these genes was confirmed in knockout and over-expression screens. Three of these genes (TPX2, TTK, and RAD54B) representing diverse pathways were further evaluated in vitro and in vivo. We demonstrate that genes identified in this study and their inhibitors (such as TTK inhibitor used in this study) have the potential to inhibit/reduce genomic instability and growth of cancer cells in vitro and in vivo. Such inhibitors also have the potential to reduce chemotherapy-induced genomic instability, while increasing their cytotoxicity.

## Results

### Integrated genomics identifies a genomic instability gene signature.
The stepwise process involved in the identification/validation of genes is presented in Fig. 1a. Briefly, the integration of copy number, expression, and survival data identified 31 genes, which were overexpressed in EAC and whose elevated expression correlated with increased genomic instability (as assessed from total copy number events) in each patient. These genes and their known roles in cancer have been shown in Supplementary Table 1.

Figure 1b shows that elevated expression of 31 gene signatures (GIS31) is associated with increased risk in The Cancer Genome Atlas (TCGA) patient dataset. Pairwise correlation also identified several subgroups of these genes showing co-overexpression. Two such examples (CDK1, TROAP, KIF4A, STIP1 and ERCC6L, MST4, NEK2) are shown by rectangles in Supplementary Fig. 1. Elevated expression of this gene signature also correlated with poor overall survival in pancreatic cancer, lung cancer, three different clinical datasets of MM, as well as in a second EAC patient dataset (GSE19417; Fig. 1c–e). Top pathways related to these genes were P53 signaling, ATP-dependent microtubule motor activity, activation of NIMA kinases, cell cycle, regulation of centriole–centriole cohesion, G2/M DNA replication checkpoint, and DNA translocase activity (Fig. 1f, g), indicating the importance of these pathways with relevance to genomic evolution.

### Functional screens confirm relevance of GIS31 genes with genomic instability.
Using EAC and MM as model systems, we have demonstrated that spontaneously elevated HR contributes to genomic instability[14], telomere length maintenance and tumor growth[17], and development of drug resistance[16]. To investigate the functional relevance of GIS31 genes, we first conducted an small interfering RNA (siRNA) screen (using validated siRNAs from Sigma) to evaluate their role in HR activity in EAC cells. Suppression of 19 out of 31 (61%) genes resulted in significant inhibition of HR activity ($p < 0.05$) (Supplementary Fig. 2). Fourteen of these genes were further evaluated in loss-of-function (using CRISPR/Cas9 system) and/or gain-of-function screens for their role in HR, micronuclei (a marker of ongoing genomic rearrangements and instability[19]), and cell viability. For the CRISPR/Cas9 loss-of-function screen, FLO-1 cells stably integrated with Cas9 were transduced with guide RNAs (two to three per gene) and the impact of gene knockdown on HR and cell viability investigated. Relative to average of two control guides, transduction with each guide RNA caused significant inhibition of HR activity (ranging from 27% to 71% inhibition; $p < 0.03$) (Fig. 2a), in three independent experiments. Although BUB1B did not show any impact in siRNA screen, its knockdown by CRISPR/Cas9 resulted in a significant inhibition of HR activity by all three guides. For all other genes, the knockdown by CRISPR/Cas9 or siRNA showed similar impact of these genes on HR, establishing their functional significance in genome maintenance. Except NEK2, CENPQ, and NUSAP1, suppression of all other genes was also associated with significant inhibition ($P < 0.05$) of cell viability (Fig. 2b). Eleven genes were overexpressed in FLO-1 cells and impact on HR activity and micronuclei (a marker of genomic instability) evaluated. Overexpression of each of these genes resulted in a significant increase (ranging from 29% to 90% increase; $P \leq 0.04$) in HR activity (Fig. 2c). Moreover, the increase in HR following overexpression correlated with its inhibition following knockdown of the same set of genes ($R^2 = 0.7$), further supporting their role in HR (Fig. 2d). Overexpression of these genes was also evaluated for impact on micronuclei (another marker of genomic instability). Figure 2e shows representative images of nuclei and micronuclei in FLO-1 cells overexpressing GIS31 genes, whereas bar graph (Fig. 2f) summarizes the results from three independent experiments. Overexpression of 6 out of 11 genes tested was associated with significant (1.8- to 3.5-fold; $p < 0.04$) increase in micronuclei (Fig. 2f). Overall 65% of identified genes had a role in one or more activities evaluated, confirming the functional relevance of identified signature to genome stability and growth.

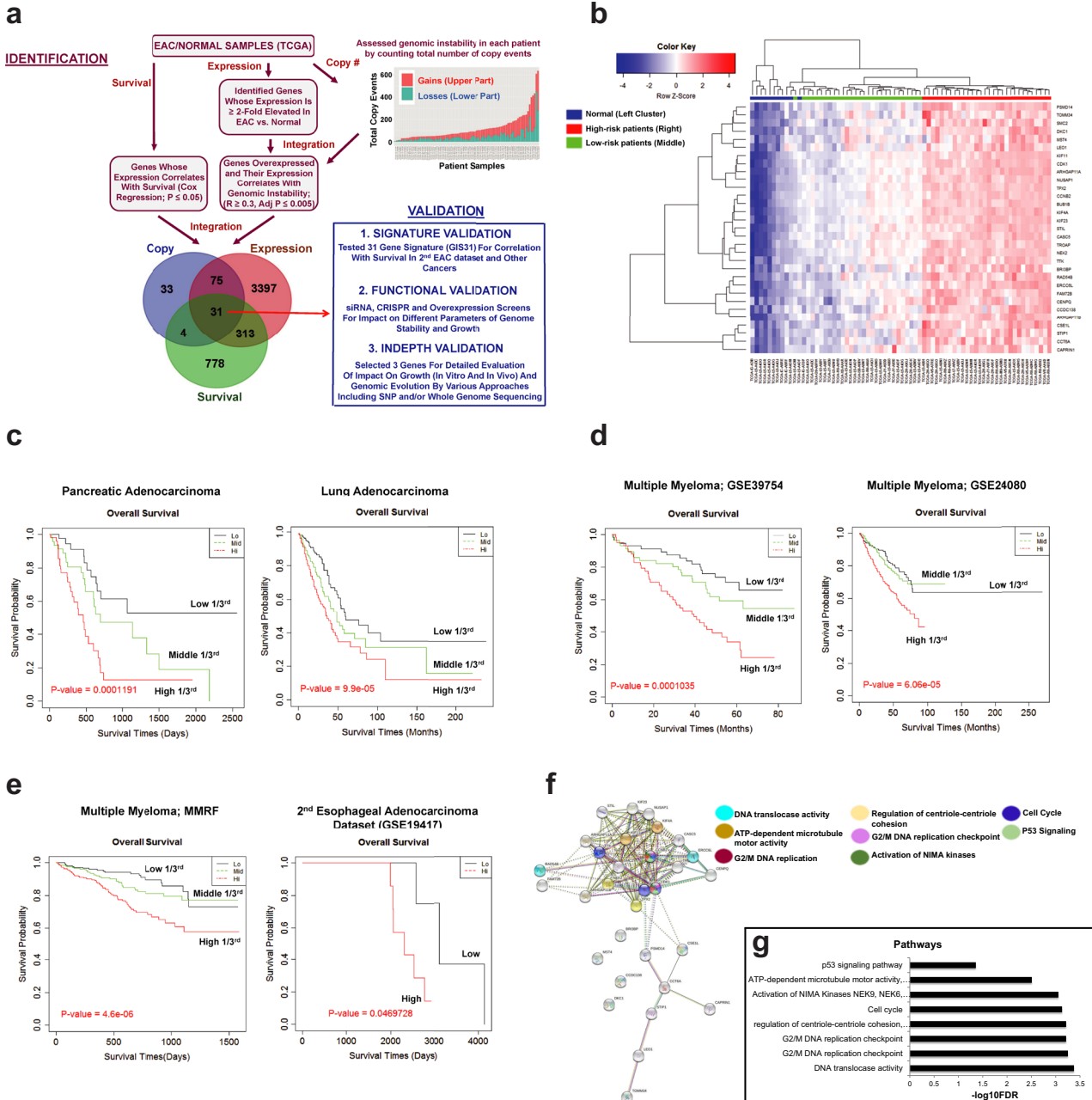

**Fig. 1 Identification of drivers of genomic evolution in EAC. a** Experimental design. Stepwise process involved in the identification and validation of genes. Identification: (1) assessed gene expression in 11 normal and 88 EAC patient samples in TCGA dataset and identified genes that were overexpressed in EAC; (2) asessed genomic instability in patient samples by counting total number of copy events in each patient. Total copy number events in each patient are shown as bar graph in Fig. 1a; each bar shows amplifications (red or upper part of bar) and deletions (green or lower part of the bar). Genomic instability data were then integrated with expression data to identify genes that were overexpressed in EAC and whose expression correlated with genomic instability; (3) evaluated resulting genes for correlation with survival. This led to identification of 31 genes that were overexpressed in EAC and whose expression correlated with genomic instability and overall survival in EAC patients in TCGA dataset. Validation: describes plan for functional validation of genes: (1) the expression of 31 gene signature was tested for correlation with survival in a second EAC dataset and in other cancers; (2) conducted siRNA, CRISPR/Cas9, and overexpression screens for impact on different parameters of genome stability and growth; (3) selected three genes for detailed evaluation of impact on genomic evolution by various approaches including SNP and/or whole genome sequencing; (4) evaluation in SCID mice. **b** Heat map shows that expression of 31 genes is high in EAC patients with poor overall survival (red bar) and low in patients with relatively better survival (green bar) (in TCGA dataset), and also low in normal samples (blue bar). **c–e** Elevated expression of 31 gene signatures (identified in TCGA patient dataset) correlated with poor overall survival in pancreatic and lung adenocarcinoma (**c**), multiple myeloma datasets GSE39754 and GSE24080 (**d**), and multiple myeloma MMRF (Multiple Myeloma Research Foundation) dataset and second esophageal adenocarcinoma dataset (GSE19417) (**e**). **f** Interconnectivity among the *GIS31* genes (based on STRING networks) is shown by lines and the shared functional roles of these genes in different pathways are indicated by different colors. **g** The pathways shown in **f** are also shown as bar graphs.

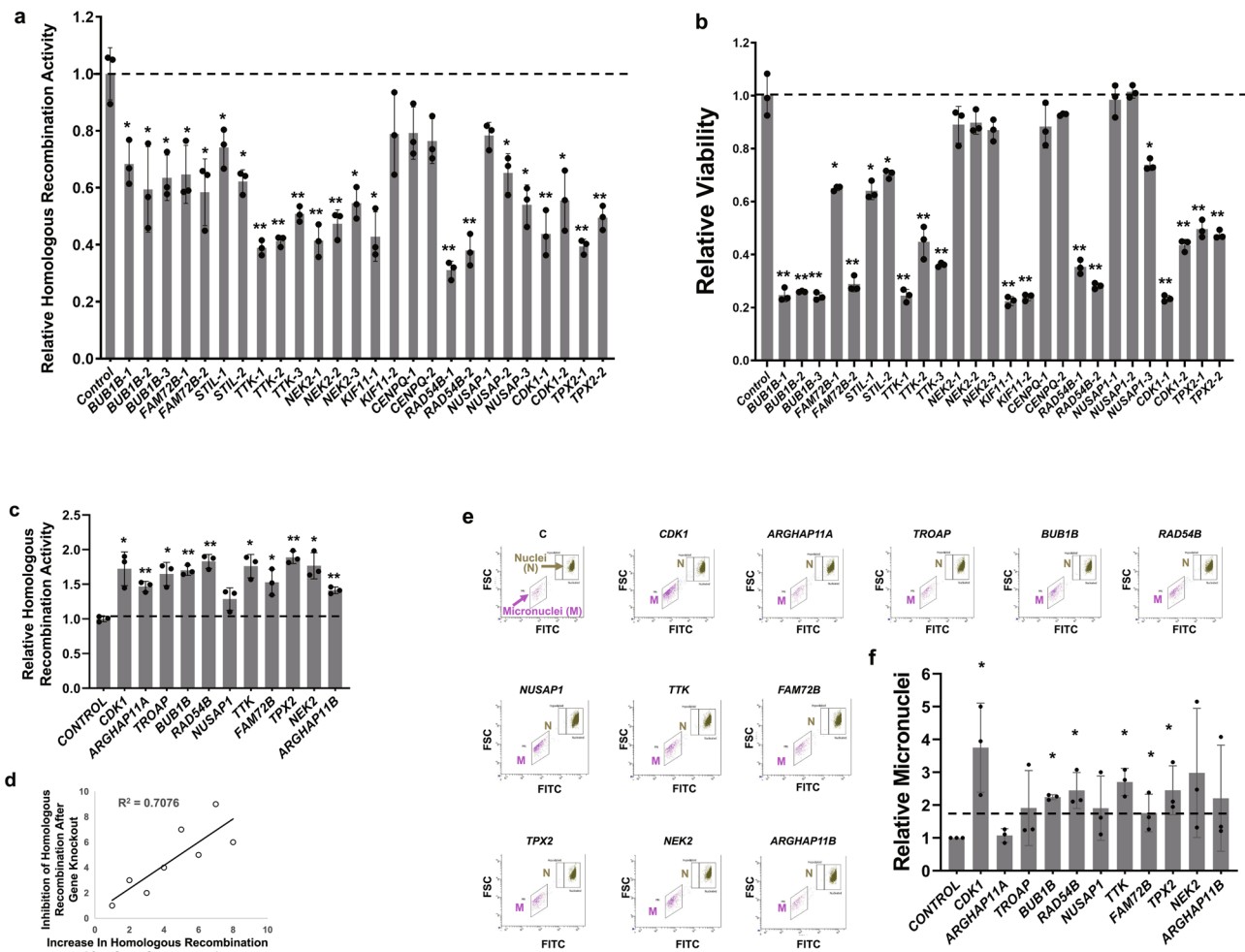

**Fig. 2 Knockdown and overexpression screens evaluating *GIS31* genes for impact on different aspects of genome stability and cell viability. a**, **b** Knockdown: EAC (FLO-1) cells, stably transduced with Cas9, were treated with lentiviral single-guide RNAs, either non-targeting control or those targeting specific genes (two or three guides per gene). Following selection, cells were cultured for 10 days and evaluated for impact on homologous recombination activity (**a**) and cell viability (**b**), relative to control (control bar represents average of two different control guides); error bars represent SDs of three experiments. Two-tailed *p*-values derived by Student's *t*-test (**p* < 0.05–0.001; ***p* < 0.001) indicate significance of difference relative to control. **c**–**f** Overexpression: FLO-1 cells were treated with lentiviral constructs, either control or those carrying open reading frames (for overexpression) of genes, selected in puromycin, and evaluated for homologous recombination (**c**) and micronuclei (a marker of genomic instability) (**e**, **f**). **c** Bar graph showing homologous recombination activity, relative to control. Error bars represent SDs of three independent experiments. Two-tailed *p*-values derived by Student's *t*-test: **p* < 0.05; ***p* < 0.0007; ****p* < 0.00005. **d** The line plot shows correlation of increase in homologous recombination following overexpression vs. its inhibition following knockdown of the same set of genes (shown in **a**). Note: for knockdown, the average inhibition of homologous recombination activity by all (two or three) guides for each gene was used. **e** Representative images showing nuclei (N) and micronuclei (MN; a marker of genomic instability) in transduced cells. *X*-axis is FITC signal indicating the amount of DNA in nuclei or micronuclei and *Y*-axis (FSC or forward scatter) indicates the size that distinguishes nuclei from micronuclei. **f** Bar graph showing fold change in micronuclei, relative to control cells. Error bars represent SDs of three experiments. *Two-tailed *p*-value derived by Student's *t*-test < 0.04.

Of genes with significant impact on these activities, *TTK* (a dual specificity protein kinase), *TPX2* (a spindle assembly factor required for normal assembly of mitotic spindles and microtubules), and *RAD54B* (a recombination protein), representing diverse pathways, were selected for further evaluation as described below.

**TTK, TPX2, and RAD54B are overexpressed in EAC cell lines and patient samples**. Expression of *TTK*, *TPX2*, and *RAD54B*, as evaluated by real-time PCR, is elevated >2-fold (*P* < 0.05) in all three EAC cell lines, relative to normal primary human esophageal epithelial cells (Fig. 3a). Error bars indicate SDs of experiments conducted in triplicate. The expression of all three genes is

also significantly elevated in EAC patients, relative to normal samples (*P* < 0.00005; Fig. 3b).

**Suppression of TTK, TPX2, and RAD54B inhibits spontaneous DNA damage and HR activity**. We suppressed *TTK*, *TPX2*, and *RAD54B* in EAC (FLO-1) cells by short hairpin RNAs (shRNAs) (Fig. 3c) and CRISPR/Cas9 system (using two guide RNAs per gene; Fig. 3g, h). Knockdown of *TTK*, *TPX2*, and *RAD54B* by shRNAs (shown in Fig. 3c) caused the inhibition of γH2AX (a DNA break marker) expression in FLO-1 cells by 51%, 77%, and 79%, respectively (Fig. 3d, e). The same blots were also evaluated for phosphorylation (on ser4 and ser8) of RPA32, a marker of end resection[20], the key step in the initiation of HR. Suppression of all three genes also led to reduction in p-RPA32 expression (by

36–77%), indicating inhibition of DNA end resection in these cells (Fig. 3d, e; full blots shown in Fig. 3d of Supplementary Data 2). Consistently, the HR activity was also significantly reduced following suppression of all three genes ($p < 0.05$; Fig. 3f). Similar observations were made following CRISPR/Cas9-mediated knockdown using two guides/gene (Fig. 3g, h). Relative to control guide RNA, each of the guide against all three genes resulted in >80% reduction in the levels of γH2AX and >70% reduction in p-RPA32 expression (Fig. 3g, h; full blots shown in Fig. 3g of Supplementary Data 2). Thus, suppression of *TTK*, *TPX2*, and *RAD54B*, whether mediated by shRNAs or CRISPR/Cas9 system, confirm their role in increased spontaneous DNA damage and dysregulation of HR in FLO-1 cells.

**Suppression of *TTK*, *TPX2*, or *RAD54B* impairs EAC cell growth in vitro and in vivo.** Control and knockdown cells were cultured and cell viability assessed at various intervals. Over a period of 7 days, the suppression of *TTK*, *TPX2*, and *RAD54B* reduced the viability of EAC cells by 54%, 39%, and 35%, respectively (Fig. 4a). To monitor the impact on growth in vivo, knockdown cells were selected in puromycin, cultured for another 2 weeks, and injected subcutaneously in severe combined immunodeficient (SCID) mice and tumor growth measured at indicated intervals. Relative to control cells, the suppression of all three genes (*TTK*, *TPX2*, and *RAD54B*) was associated with significant reduction in tumor size ($p \leq 0.04$) in SCID mice (Fig. 4b–d).

**Transgenic overexpression of *TTK*, *TPX2*, and *RAD54B* in normal esophageal cells increases spontaneous DNA damage and HR activity.** Expression of *TTK*, *TPX2*, and *RAD54B* was upregulated in normal primary human esophageal epithelial cells, using lentivirus-based expression plasmids. Overexpression of all three genes (Fig. 5a) led to an increase in the expression of γ-H2AX (DNA break marker), p-RPA32 (marker of DNA end resection) (Fig. 5b; full blots shown in Fig. 5b of Supplementary Data 2), as well as significant increase in HR activity ($p < 0.001$; Fig. 5c) in normal esophageal cells. These data are consistent with knockdown experiments (shown in Fig. 3), which showed that suppression of these genes in EAC cells inhibits DNA breaks and HR activity. Moreover, the overexpression of all three genes in EAC (OE19) cell also led to an increase in the spontaneous DNA breaks and DNA end resection, as well as a significant increase in HR activity (Supplementary Fig. 3a, b). Consistently, the overexpression of all three genes in EAC (FLO-1) cells also caused an increase in spontaneous DNA breaks and DNA end resection, a distinct step in the initiation of HR (Supplementary Fig. 4a); the overexpression of these genes was confirmed by quantitative PCR (Supplementary Fig. 4b). We also suppressed these genes in normal cells (fibroblasts). However, as expected from a relatively intact genome, no impact of these suppressions was detected on DNA breaks or genome stability. Thus, the transgenic overexpression in both the normal cell types (with low expression of these genes) and EAC cell lines (with high expression of these genes), as well as the suppression in EAC cell lines confirm that the elevated expression of *TTK*, *TPX2*, and *RAD54B* contributes to increased spontaneous DNA breaks and HR activity.

**WGS confirms the role of identified genes in genomic instability and provides evidence of underlying mechanism/s.** Normal primary human esophageal epithelial cells transduced with control plasmid or those overexpressing *TTK*, *TPX2*, or *RAD54B* (shown in Fig. 5a) were cultured for 60 days. DNA from these and "day 0" cells (representing baseline genome) was purified and analyzed by whole genome sequencing (WGS; 30×).

Genome of day 0 cells was used as baseline to identify new genomic changes acquired by transduced cells over a period of 60 days. Circos plots in Fig. 5d show new copy number events in control (outer circle) and transgene-overexpressing cells (inner circle), relative to day 0 cells. The bar graph (Fig. 5e) summarizes total copy number segments over whole genome in these cells. Relative to baseline genome (of day 0 cells), the control plasmid-transduced cells acquired 53 events in 60 days, whereas overexpression of *TPX2*, *RAD54B*, and *TTK* led to acquisition of 614, 1026, and 729 events, respectively (Fig. 5d, e). This shows that the overexpression of these genes caused a 12- to 19-fold increase in copy number events over time, indicating a marked increase in genomic instability.

Normal primary human esophageal epithelial cells transduced with control plasmid or those overexpressing *TTK*, *TPX2*, or *RAD54B* (shown in Fig. 4a) were also evaluated by SNP (SNP6.0) arrays (Affymetrix), at an earlier time point (day 30). Again, the genome of "day 0" cells was used as baseline to identify new genomic changes acquired by transduced cells over a period of 30 days. Relative to baseline genome, the overexpression of all three genes was associated with the acquisition of new copy number events throughout chromosomes (Supplementary Fig. 5a), ranging from 4.4- to 4.9-fold increase relative to control shRNA-transduced cells (Supplementary Fig. 5b). Thus, evaluation by WGS (at day 60) and evaluation by SNP arrays (at day 30) demonstrated that overexpression of these genes increases genomic instability in normal esophageal cells. Impact of the overexpression of these genes in EAC (OE19) cells was also evaluated for micronuclei, a marker of genomic instability. Overexpression of all three genes was associated with significant increase ($p < 0.05$) in the percentage of micronuclei in the cells (Supplementary Fig. 6), indicating increased genomic instability.

WGS data were also analyzed for mutations. Overexpression of *TPX2*, *TTK*, and *RAD54B* increased the mutational load in normal esophageal cells by 1.37-fold (>8000 new mutations), 1.46-fold, and 1.87-fold, respectively (Fig. 5f). The somatic mutations in a cancer genome bear specific signatures or scar marks of distinct mutational processes, or the mechanisms that give rise to these mutations. Considering all possible combinations of substitutions in a trinucleotide context, specific mutational signatures (indicative of underlying mutational processes) have been identified[21]. Using previously published method[22,23], we extracted the mutational signatures activated by overexpression of *TPX2*, *RAD54B*, and *TTK* in normal esophageal cells. By using previously published esophageal cancer WGS dataset as the reference[24], we deconvolved the contributing mutational signatures to each sample. Signature #3 (DNA double-strand break) and reference Signature 5 (of unknown etiology) covered the majority of the newly acquired mutations in the three experimental samples compared to control sample (Fig. 5g). This is consistent with loss- and gain-of-function studies showing the role of these genes in increased spontaneous DNA breaks and HR activity (a repair mechanism for DNA double-strand breaks).

Thus, WGS, SNP, and other functional data confirmed that the overexpression of these genes caused a massive genomic instability, enabling these cells to acquire a variety of genomic changes during growth in culture.

**A small-molecule inhibitor of TTK inhibits EAC cell growth in vitro and in vivo, and increases efficacy of chemotherapeutic agents.** Normal or non-cancerous cell types (HEsEpi, primary human esophageal epithelial cells; Het-1A, SV40 large T-antigen-transfected human esophageal epithelial cells; and HDF, human diploid fibroblasts) and EAC cell lines (FLO-1, OE19, and OE33)

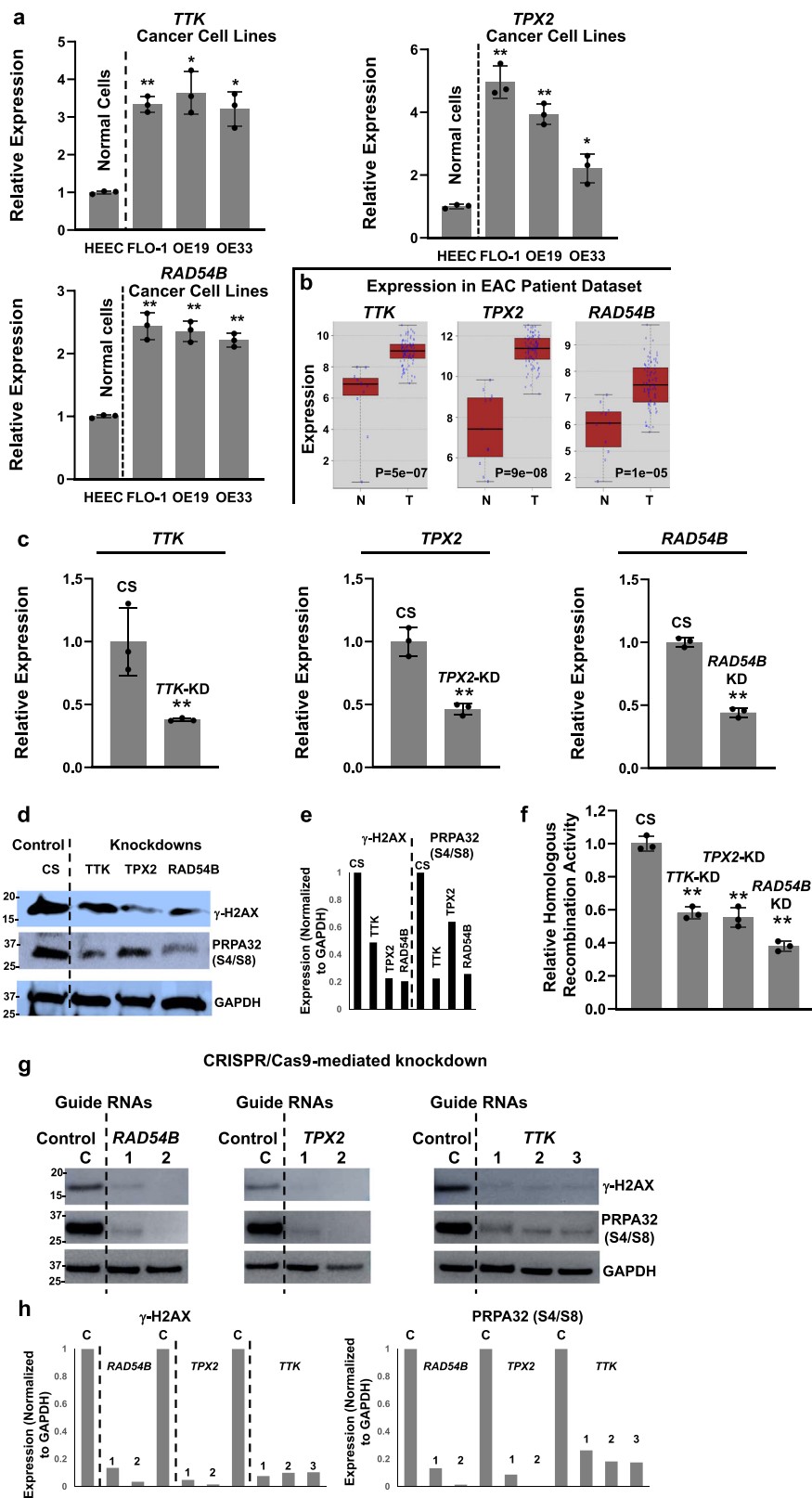

were treated with TTK inhibitor at various concentrations for 72 h and cell viability assessed. IC50 values of the inhibitor was much higher for non-cancerous cells (HEsEpi, 252 μM; Het-1A, 5.5 μM; HDF, 3.6 μM), whereas 1.0 μM for all three EAC cell lines (Fig. 6a). To evaluate the efficacy of TTK inhibitor in vivo, FLO-1 cells were injected subcutaneously in SCID mice and, following the appearance of palpable tumors, mice treated with the drug.

Average tumor size in treated mice was significantly smaller than that in control mice ($p = 0.01$; Fig. 6b).

To evaluate the impact of TTK inhibitor on efficacy of chemotherapeutic agents, EAC cell lines (FLO-1 and OE19) were treated with inhibitor, alone as well as in the presence of 5-fluorouracil or cisplatin, and cell viability measured after 72 h. TTK inhibitor increased cytotoxicity of both the fluorouracil

**Fig. 3 Suppression of elevated *TTK, TPX2,* or *RAD54B* expression in EAC cells inhibits spontaneous DNA breaks, homologous recombination, and growth in vitro and in vivo. a** Gene expression, evaluated in normal primary human esophageal epithelial (HEsEpi) cell and EAC cell lines (FLO-1, OE19, and OE33), using real-time PCR. Error bars indicate SDs of experiments conducted in triplicate. Two-tailed *p*-values derived by Student's *t*-test: *$p < 0.05$–0.001; **$p < 0.001$. **b** Log2 gene expression in EAC relative to average expression in normal samples in TCGA patient dataset ($n = 88$). **c–f** shRNA-mediated suppression of *TTK, TPX2,* or *RAD54B* reduces spontaneous DNA breaks and homologous recombination in EAC cells. FLO-1 cells were transduced with lentivirus-based shRNAs, either control (CS) or those targeting *TTK, TPX2, RAD54B,* and following selection, knockdown (KD) of gene expression confirmed by RT PCR (**c**), levels of γ-H2AX and phosphorylated-RPA32 evaluated by western blotting (**d**, **e**), and homologous recombination activity measured using a functional assay (**f**). Error bars in **c** indicate SDs of experiment conducted in triplicate and those in **f** are SDs of three experiments. Two-tailed *p*-values derived by Student's *t*-test: *$p < 0.05$–0.001; **$p < 0.001$. Western blot image (**d**) and quantification of the blot (**e**) are shown. **g,h** CRISPR/Cas9-mediated suppression of *TTK, TPX2,* or *RAD54B* reduces spontaneous DNA breaks and DNA end resection in EAC cells. FLO-1 cells, stably transduced with Cas9, were treated with lentiviral single-guide RNAs, either non-targeting control (C) or those targeting these genes (two or three guides per gene). Following selection in puromycin, cells were cultured for 10 days and impact of their suppression on levels of γ-H2AX (a marker of DNA breaks) and p-RPA32 (a marker of DNA end resection) evaluated by western blotting. Images (**g**) and the bar graphs presenting quantification of western blots (**h**) are shown.

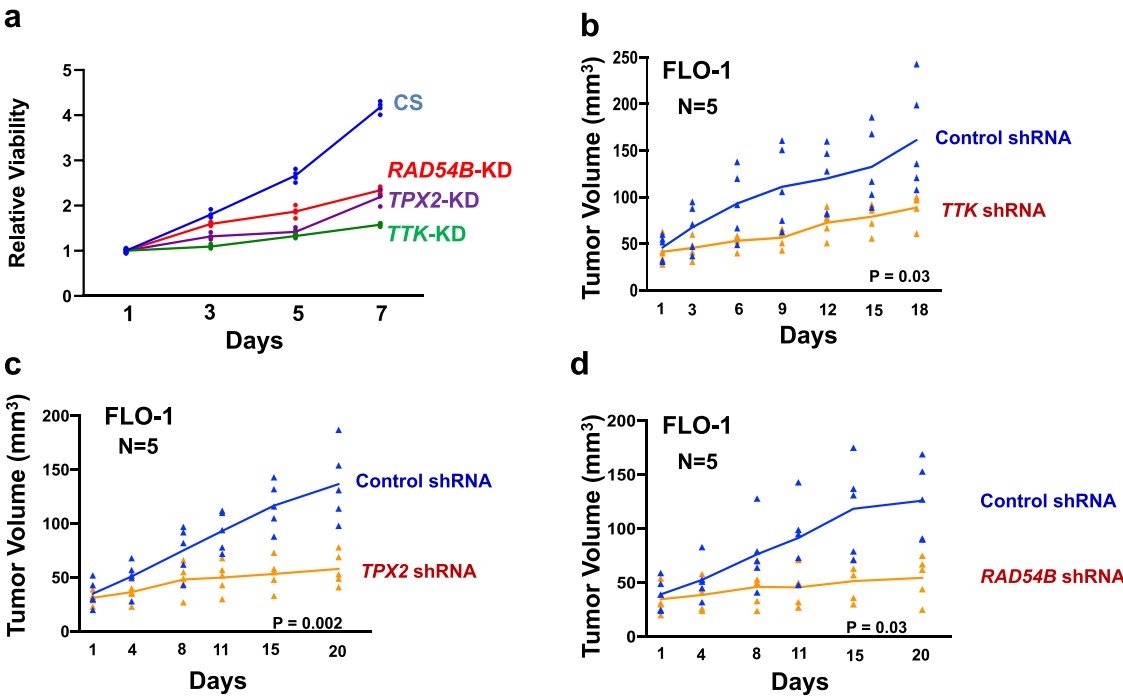

**Fig. 4 Suppression of *TTK, TPX2,* or *RAD54B* impairs EAC cell growth in vitro and in vivo. a** Control and knockdown FLO-1 cells were cultured and cell viability assessed at various intervals. **b–d** Control and knockdown FLO-1 cells, cultured for >10 days in vitro, were subcutaneously injected in SCID mice and tumor growth measured at indicated intervals. Error bars indicate SEMs of tumor sizes in different mice. Two-tailed *p*-values derived by Student's *t*-test ≤ 0.04.

(Fig. 6c) and cisplatin (Fig. 6d). Combination index plots (Fig. 6e) show that TTK inhibitor synergistically increased the cytotoxicity of both chemotherapeutic agents in both cell lines tested. In summary, these data show that TTK inhibitor inhibits EAC cell growth both in vitro and in vivo, and synergistically increases the efficacy of chemotherapeutic agents.

**Small-molecule inhibitor of TTK inhibits spontaneous DNA damage and HR activity, and reverses genomic instability caused by chemotherapeutic agent, in EAC cells.** EAC cell lines (FLO-1 and OE19) were treated with low doses of TTK inhibitor, alone or in the presence of chemotherapeutic agent, and substrate-attached (live) cells evaluated for various parameters of genome stability. TTK inhibitor reduced HR activity in EAC cell lines in a dose-dependent manner (Fig. 7a). DNA breaks (as assessed from γ-H2AX expression) were reduced by TTK inhibitor, whereas they were increased (>10-fold) by etoposide (Fig. 7b; full blots shown in Fig. 7b of Supplementary Data 2).

Importantly, the addition of TTK inhibitor caused a marked (fivefold) inhibition/reduction of etoposide-induced DNA breaks and DNA end resection (as assessed from p-RPA32 expression) (Fig. 7b; full blots shown in Fig. 7b of Supplementary Data 2). EAC cell lines treated with TTK inhibitor and/or etoposide were also evaluated for impact on micronuclei, a marker of genomic instability. In FLO-1 cells, etoposide caused an increase in the micronuclei (by 12-fold), whereas a low dose of TTK inhibitor caused ~3.0-fold reduction in etoposide-induced micronuclei (Fig. 7c, d). Similarly, in OE19 cells, etoposide caused an increase in micronuclei (by 5.7-fold), whereas treatment with a low dose of TTK inhibitor not only inhibited spontaneous micronuclei (by 2-fold) but also caused 2.6-fold reduction in etoposide-induced micronuclei (Fig. 7e, f). A similar impact of these treatments was also observed on acquisition of copy number events in EAC (FLO-1) cells. FLO-1 cells were treated with TTK inhibitor, etoposide, and a combination of both drugs for 3 weeks, and the acquisition of copy number changes relative to "Day 0" cells (representing baseline genome) monitored, using Axiom™

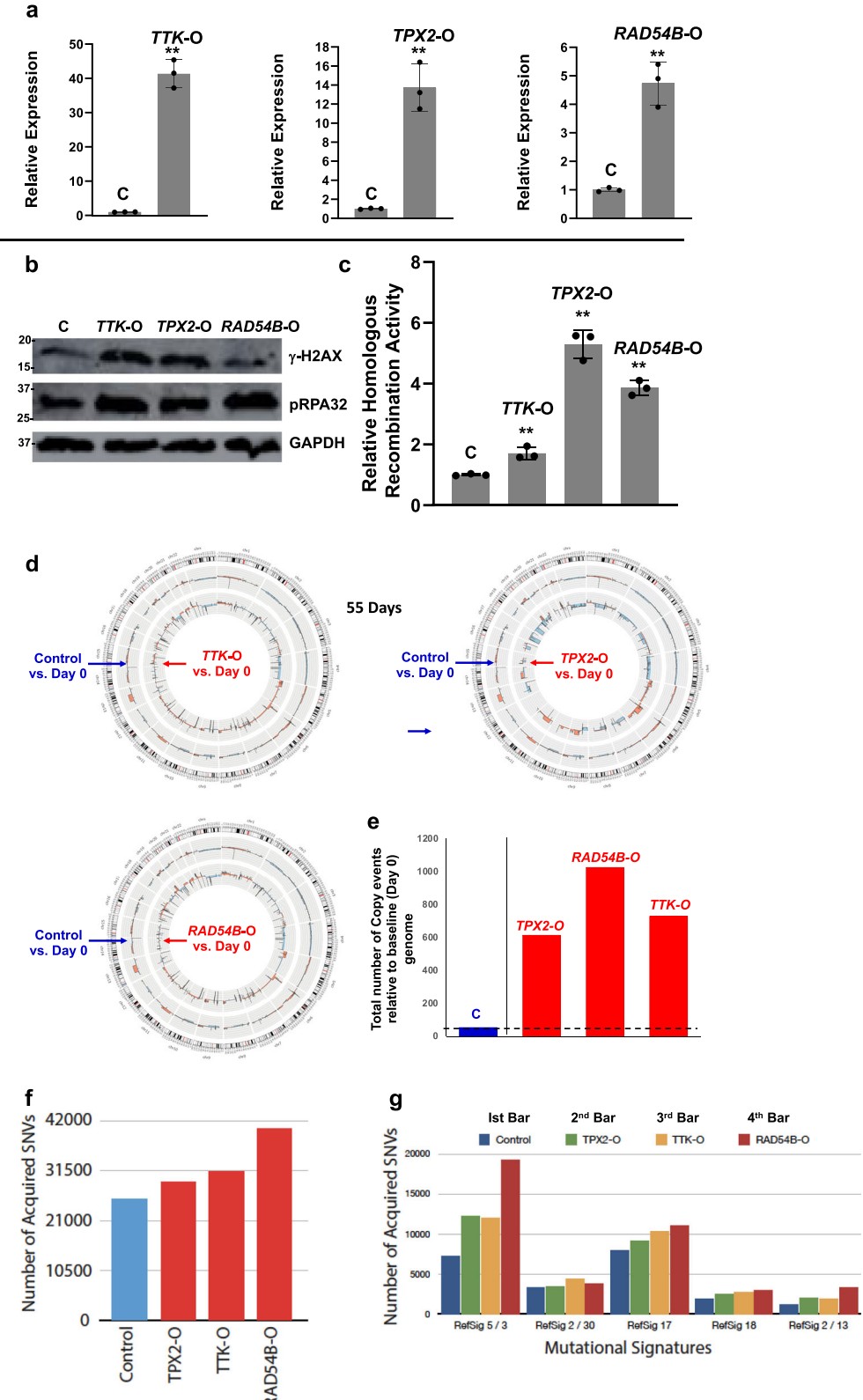

Precision Medicine Diversity Arrays. Etoposide increased the acquisition of copy number events by >2-fold relative to control cells, whereas addition of TTK inhibitor was able to reduce/prevent this increase by 65% (Supplementary Fig. 7). Taken together, these data indicate that TTK inhibitor can potentially reduce spontaneous and chemotherapeutic agent-induced DNA damage and genomic instability in EAC cells.

## Discussion

Genomic instability enables a cell to constantly acquire genomic changes, some of which underlie the initiation and/or progression of cancer to advanced stages of disease. Similar to most other cancers, a marked genomic instability has also been observed in EAC and its premalignant states[3,4,9,10]. Consistently, the EAC genome is highly aberrant[9] and the cancer is heterogeneous and

**Fig. 5 Overexpression of _TTK_, _TPX2_, or _RAD54B_ activates mechanisms leading to genomic instability in normal esophageal cells.** Normal primary human esophageal epithelial cells (HEsEpi) were transfected with control plasmid (C) or those overexpressing _TTK_ (_TTK-O_), _TPX2_ (_TPX2-O_), or _RAD54B_ (_RAD54B-O_), selected in puromycin, and evaluated for various aspects of genome stability at indicated time points. **a**–**c** Impact on DNA breaks and homologous recombination activity. At day 7 after transfection, the transgene overexpression was confirmed by Q-PCR (**a**) and cells evaluated for γ-H2AX and phosphorylated-RPA32, using western blotting (**b**), and homologous recombination activity, using a plasmid-based assay (**c**). Error bars indicate SDs of experiments conducted in triplicate. Two-tailed _p_-values derived by Student's _t_-test: **_p_ < 0.001. **d**–**g** Impact on genomic instability assessed using whole genome sequencing (WGS). **d** Circos plots show acquisition of copy number events in control (outer circle) and transgene-overexpressing cells (inner circle) acquired in 60 days, relative to baseline genome of day 0 cells. **e** Bar graph showing total copy number segments (amplifications and deletions) throughout genome in these cells. **f**, **g** Mutational instability and underlying processes. WGS data shown in **b** were also analyzed for mutations and mutational processes. **f** Total number of newly acquired mutations relative to parental genome (day 0 sample); the mutations found in "day 0 sample" were removed from all samples. **g** Representation of mutational processes extracted in all four samples. For each sample, the contribution of the mutational signatures were calculated using signal (https://signal.mutationalsignatures.com) with esophageal dataset[24] as the reference set. For each sample (color coded bars), the number of mutations (_y_-axis) contributed by mutational signatures (_x_-axis) are shown.

chemoresistant[11]. Therefore, identification of genes, which drive genomic evolution, can help develop effective strategies for prevention and treatment of EAC.

Using EAC and MM as model systems, we previously showed that dysregulated HR significantly contributes to genomic instability[14,16], development of drug resistance[16], and tumor growth[17]. Here we used an integrated genomics approach to identify drivers of genomic evolution in EAC. As genomic instability is associated with cancer progression and some of the mechanisms of genomic instability (such as HR) are also required for tumor growth, we hypothesized that if elevated expression of a gene correlates with increased genomic instability and poor survival in EAC patients, it could be a potential driver of genomic evolution. Although downregulation of a gene could also mediate genomic instability, in this study we focused only on those whose overexpression correlated with genomic instability. This is because it is relatively easy to target a gene that is overexpressed. Integration of expression, copy number and survival information (in TCGA dataset) identified 31 gene signature whose elevated expression correlated with poor survival in a second (different) EAC dataset, as well as in pancreatic and lung cancers, and also in three different MM datasets. This indicates that the signature or at least a subset of its genes are involved in genomic instability in other cancers as well. For subsequent validation, all 31 genes were first screened (using validated siRNAs) for impact on HR activity. Subsequently, 14 of these genes were further evaluated in CRISPR/Cas9-mediated loss-of-function and/or overexpression screens for their role in HR and/or micronuclei (a marker of genomic instability[19]), as well as cell viability. Initial siRNA screen (evaluating all 31 genes), demonstrated that 61% of these genes had a functional role in dysregulation of HR activity in EAC cells. Overall, the siRNA, CRISPR/Cas9 knockdown, and overexpression screens indicated that majority of the genes identified by integrated genomics had a functional role in genomic instability and growth of EAC cells.

Three of these genes, which had a significant impact on the activities tested, had >4-fold higher expression in EAC relative to normal tissues (in TCGA dataset), and represented diverse pathways (TTK, a dual specificity protein kinase; TPX2, a spindle assembly factor required for normal assembly of mitotic spindles and microtubules; and RAD54B, a recombination protein), were selected for further evaluation. Suppression of all three genes reduced DNA breaks and HR activity in EAC cells. The process of HR is initiated by exonucleolytic resection resulting in the formation of a 3′ single-stranded DNA overhang, which is immediately quoted by phosphorylated-RPA32 (pRPA). Thus, pRPA serves as DNA end resection marker[20,25]. Suppression of all three genes led to inhibition of DNA end resection, suggesting that these genes contribute to HR at some early step.

RAD54B is a known HR protein[26] and its elevated expression is associated with poor prognosis in colorectal[27] and lung[28] cancers. Consistently, our data in EAC show that elevated RAD54B expression contributes to increased HR activity and genomic instability, and correlates with poor survival. Role of TPX2, a microtubule-associated protein, was also confirmed in HR in EAC cells. _TPX2_ has been shown to contribute to survival in ovarian[29] and endometrial[30] cancers. Cancer cells with increased genetic instability also show increased sensitivity to suppression of _TPX2_[31]. Our data demonstrate a role of _TPX2_ in HR in EAC. Moreover, the evaluation by both the SNP arrays and by WGS shows that overexpression of _TPX2_ induces genomic instability in normal esophageal cells. This is consistent with the association of TPX2 with progression in prostate cancer patients[32]. Finally, the evaluation of TTK, the spindle assembly checkpoint kinase, also confirmed its role in dysregulation of HR and genome stability in EAC. This is also consistent with a recent report demonstrating the role of _TTK_ in HR in breast cancer[33]. _TTK_ has been identified as one of the promising candidates for vaccination in esophageal cancer patients with advanced stage disease[34,35]. Role of _TTK_ in survival of cancer cells and cytotoxicity of its inhibitors has also been demonstrated in hepatocellular carcinoma[36] and glioblastoma[37].

Although TPX2, TTK, and RAD54B seem to represent diverse biological pathways, the overexpression of all three genes was associated with increased DNA damage, HR activity, and genomic instability. This is also consistent with the signatures of mutational processes extracted from the WGS data. Signature 3 (DNA double-strand break) and Signature 9 (_AID_ or activation-induced cytidine deaminase, a gene that induces somatic hypermutation and class-switch recombination) were among the top signatures covering a majority of mutational landscape. Features of Signature 3 (https://cancer.sanger.ac.uk/cosmic/signatures_v2, including loss of repair of DNA double-strand breaks by HR and increase in large deletion/insertion events, which exhibit a microhomology at their breakpoint sites) suggest that it indicates aberrant/dysregulated HR. Consistently, our loss- and gain-of-function studies also confirm that overexpression of these genes contributes to increased HR activity. Similarly, Signature 9 indicates activity of _AID_, a gene that creates mutations in DNA by deamination of cytosine to uracil and contributes to somatic hypermutation and class-switch recombination. It is possible that AID-induced DNA damage also contributes to increased HR. Our functional screens, subsequent loss- and gain-of-function studies, and previous investigations in EAC[14,15] suggest that dysregulated HR is probably a common or predominant mechanism that drives genomic instability in EAC and possibly in other cancers[16]. Evaluation of structural variations in human genomes has identified non-allelic (aberrant) HR as one of the top underlying mechanisms[38]. Although HR is the most precise DNA repair

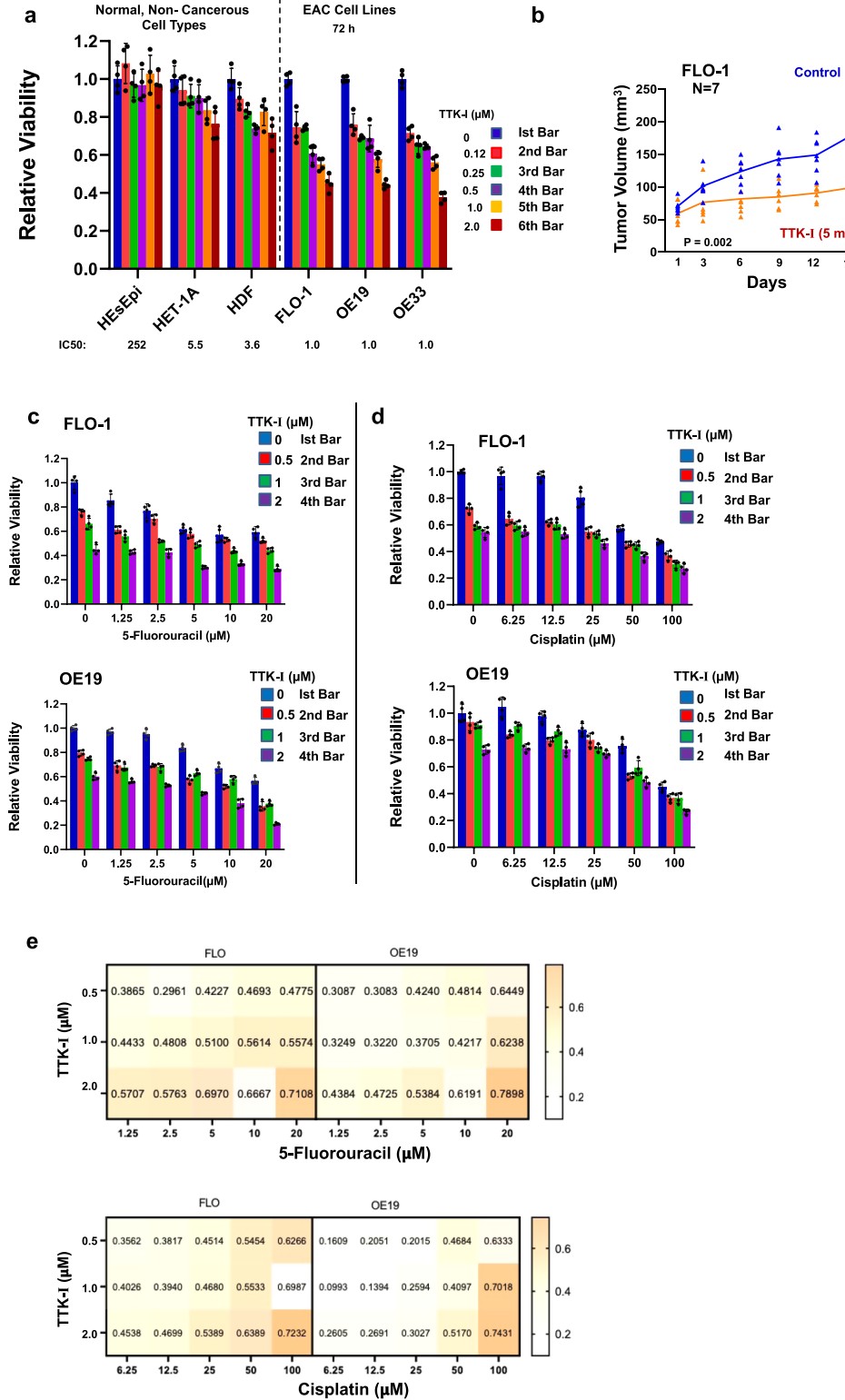

**Fig. 6 A small-molecule inhibitor of TTK inhibits EAC cell growth in vitro and in vivo, and increases efficacy of chemotherapeutic agents. a** Normal or non-cancerous cell types (HEsEpi, primary human esophageal epithelial cells; Het-1A, SV40 large T-antigen-transfected human esophageal epithelial cells; HDF, human diploid fibroblasts) and EAC cell lines (FLO-1, OE19, AND OE33) were treated with TTK inhibitor (TTK-I, CFI402257) at various concentrations for 72 h and cell viability assessed. Error bars represent SDs of three experiments. **b** EAC (FLO-1) cells were injected subcutaneously in SCID mice and, following the appearance of palpable tumors, mice treated with TTK inhibitor. Tumor volumes were measured at indicated intervals. Error bars represent SDs of tumor sizes from nine control and nine treated mice. **c–e** EAC cell lines (FLO-1 and OE19) were treated with inhibitor (TTK-I, CFI402257), alone as well as in the presence of chemotherapeutic agents 5-fluorouracil (**c**) or cisplatin (**d**), and cell viability measured after 72 h. Error bars represent SDs of three experiments. Panel **e** shows combination indexes (calculated using CalcuSyn software) of TTK-I with chemotherapeutic agents 5-fluorouracil or cisplatin; a value <1 indicates a synergistic impact.

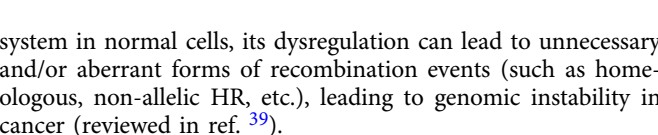

system in normal cells, its dysregulation can lead to unnecessary and/or aberrant forms of recombination events (such as homologous, non-allelic HR, etc.), leading to genomic instability in cancer (reviewed in ref. [39]).

Evidence from pancreatic and breast cancers suggests that the mechanism of cell death following treatment with TTK inhibitors is induction of genomic instability[36,40]. This seems to be in contradiction with our results in EAC showing that treatment with TTK inhibitor reduces spontaneous DNA damage and genomic instability. However, it should be noted that we removed dead cells and evaluated the genomic impact of TTK inhibitor in cells that survived the treatment. This is extremely important, because cell death/apoptosis is also associated with DNA breaks, which can hinder/confuse the results during evaluation of the

**Fig. 7 Small-molecule inhibitor of TTK inhibits spontaneous homologous recombination activity and DNA breaks, and reverses genomic instability caused by chemotherapeutic agent, in EAC cells. a** TTK inhibitor inhibits homologous recombination activity in EAC cells. FLO-1 cells were treated with different concentrations of TTK inhibitor (TTK-I, CFI402257) for 48 h and impact on homologous recombination activity evaluated using a plasmid-based assay. Error bars indicate SDs of experiment conducted in triplicate. Two-tailed $p$-values derived by Student's $t$-test: **$p < 0.001$. **b** TTK inhibitor reduces etoposide-induced DNA breaks and end resection in EAC cells. FLO-1 cells were treated with TTK inhibitor (TTK-I) and etoposide (ET), alone as well as in combination with each other (as indicated) for 48 h, and impact on levels of $\gamma$-H2AX, PARP, and phosphorylated-RPA32 evaluated by western blotting. **c–f** TTK inhibitor reverses etoposide-induced genomic instability in EAC cells. EAC cells, FLO-1 (**c**, **d**) and OE19 (**e**, **f**), were treated with TTK inhibitor (TTK-I) and etoposide (ET), alone as well as in combination, as described in **b** and impact on micronuclei (a marker of genomic instability) assessed. Flow cytometry images of nuclei (N) and micronuclei (M) (**c**, **e**) and bar graphs showing percentage of micronuclei (**e**, **f**) are shown.

impact of treatment on genomic integrity/stability. This is also the reason that we had to use lower doses of TTK inhibitor (where cell death is minimum) to accurately determine its impact on genomic integrity and stability. Treatment with TTK inhibitor at low doses inhibited both the spontaneous DNA breaks and genomic instability in EAC cells. This is consistent with our other observations showing that transgenic suppression of *TTK* in EAC cells inhibits spontaneous DNA damage and HR activity, whereas transgenic overexpression of *TTK* in normal esophageal epithelial cells increases spontaneous DNA damage, HR activity, and genomic instability, as assessed by whole-genome SNP arrays and sequencing. However, the difference between our data and that from other cancers[36,40] could also be attributed to different mechanisms operative in these cancers. Moreover, the role of *TTK* in prostate cancer progression[41] is also consistent with our data, demonstrating that *TTK* overexpression dysregulates HR and genome stability in EAC cells.

Knockdown screen indicated that the genes identified as potential drivers of genomic evolution were also involved in EAC cell viability. Moreover, transgenic suppression of *TTK*, *TPX2*, and *RAD54B* not only inhibited EAC cell growth in vitro but also significantly inhibited their growth as tumors in SCID mice. The treatment of EAC cells with a TTK inhibitor impaired their growth and synergistically increased cytotoxicity of chemotherapeutic agents. Importantly, the inhibitor of TTK inhibited spontaneous and chemotherapy-induced genomic instability in EAC cells. This seems to be in contrast to a general notion that genomic chaos must be enhanced (not diminished) to kill cancer cells. This is certainly true for chemotherapeutic agents, as they mostly kill cancer cells by inducing DNA damage. However, this is not an ideal approach of killing cancer cells, because the treatment with DNA-damaging agents not only disrupts genomic integrity of normal cells but also increases DNA damage in surviving cancer cells, thus predisposing them to increased genomic instability and its harmful impact. Alternatively, the genes involved in genomic instability and growth of cancer cells (such as *TTK* and those identified here) can be targeted to initiate growth arrest in cancer cells without first increasing genomic instability or chaos. When such inhibitors are combined with chemotherapy, the chemotherapy-induced cytotoxicity increases, whereas genomic instability is reduced. This is because when cancer cells (with already increased DNA breaks) are subjected to chemotherapy (such as etoposide), a large fraction of cells acquire a very high level of DNA breaks and are killed. However, there is also a fraction of cells that acquire a small or moderate increase in DNA breaks. The increase in DNA breaks in these cells is also associated with increase in HR activity, which not only helps in their survival (by reducing the number of DNA breaks) but also contributes to increase in genomic instability by (utilizing some of these breaks in) unnecessary/aberrant recombination events. When etoposide is combined with TTK inhibitor, the HR activity is reduced. This results in an increase in the fraction of cells with extensive DNA breaks (because of reduced HR), leading to increased cell death. However, the reduced HR activity in these

surviving cells results in a more stable genome, as demonstrated in Fig. 7. Consistently, we have demonstrated that inhibition of HR in EAC cells inhibits genomic instability[14] and tumor growth[17]. Our recent data in MM have also demonstrated that inhibition of *APEX1* nuclease, a gene that drives genomic evolution in myeloma, increases cytotoxicity while inhibiting genomic instability caused by a chemotherapeutic agent[42].

The integrated genomics approach used here can identify drivers of genomic evolution. *RAD54B*, *TTK*, and *TPX2* are identified as genes involved in dysregulation of HR and genome stability in EAC. Small-molecule inhibitors of TTK and other genes identified in this study have the potential to inhibit/delay genomic evolution and tumor growth. Such inhibitors also have the potential to increase cytotoxicity, while reducing harmful genomic impact of chemotherapy.

## Methods

**Integrated genomics.** EAC patient dataset ($n = 88$) from TCGA was used. We hypothesized that if elevated expression of a gene correlates with increased genomic instability and poor survival of a patient, it could be a potential driver of genomic evolution. To identify these genes, we used the following stepwise process; details of each step are provided below: (1) investigated gene expression in 11 normal and 88 EAC patient samples in TCGA dataset and identified genes that were overexpressed in EAC; (2) assessed genomic instability in patient samples by counting the total number of copy events in each patient. Integrated genomic instability data with expression data to identify genes that were overexpressed in EAC and whose expression correlated with genomic instability; (3) evaluated resulting list of genes for correlation with survival. This led to the identification of 31 genes that were overexpressed in EAC and whose expression correlated with genomic instability and overall survival in EAC patients in TCGA dataset; (4) the expression of this 31 gene signature was tested for correlation with survival in a second EAC dataset and in other cancers.

RNASeq raw counts of TCGA data (containing 95 cases for esophageal squamous cell carcinomas, 88 cases of EACs, and 11 normal samples: Level 3, Illumina Hiseq RNASeqV2) were downloaded and pre-processed separately, using the package TCGAbiolinks[43]. Each subtype expression matrix was normalized using the TCGAanalyze_Normalization function encompassing the EDASeq protocol[24]. Differential expression was computed with the TCGAanalyze_DEA function, implementing the EdgeR protocol[44]. A cutoff of 1 was set for log2-fold change (in EAC relative to average of normal samples) with a false discovery rate (FDR) of 0.01, to identify genes overexpressed in EAC. Only EAC data were subsequently used in this study.

Genomic instability in each patient was assessed from the total number of copy number events, which were computed using an R script. A copy event was defined as "high" if the segment mean was ≥2.5 and "low" if the segment mean was ≤1.5. Overall survival was defined as the day of last follow-up or day of death depending on vital status. Samples with a value of "NA" in the vital status row were excluded. Univariate analysis was performed using Cox regression with gene expression as the independent variable. For each gene, we recorded regression coefficient and hazard ratio with $P$-value cutoff of 0.05.

**Functional screens to validate genomic instability signature.** Expression of 31 genes, identified as potential drivers of genomic instability in EAC (Fig. 1), was either suppressed (using validated siRNAs and CRISPR/Cas9 technology) and/or overexpressed (using open reading frame expression plasmids in viral constructs) in EAC (FLO-1) cells and impact on micronuclei (marker of genomic instability), HR (a mechanism of genomic instability in EAC), and cell viability assessed.

**Constructs for loss- and gain-of-function, and transductions/transfections.** Validated siRNAs and lentiviral constructs expressing shRNAs were purchased from Sigma. Constructs for CRISPR/Cas9-mediated gene knockout (Cas9 and

guide RNAs) and plasmids expressing open reading frames (for overexpression) were obtained from Dr. David Root at Broad Institute, Boston, MA. For CRISPR/Cas9-mediated gene knockout, EAC cells were stably transduced with Cas9 and then transduced with guide RNAs targeting individual genes. For shRNA-mediated gene suppression/overexpression, lentiviral constructs were transduced in normal and EAC cells as reported previously[14,15,17].

**Cell types.** Normal primary human esophageal epithelial (HEsEpiC) cells were purchased from ScienCell Research Laboratories (Carlsbad, CA); SV40 large T-antigen-transfected human esophageal epithelial cells (Het-1A; CRL-2692) and normal human diploid fibroblasts were purchased from American Type Culture Collection (Manassas, VA); and EAC cell lines (FLO-1, OE19, and OE33) were purchased from Sigma Aldrich Corporation (Saint Louis, MO). Cells were cultured as reported previously[14,15,17,45].

**Antibodies and reagents.** Anti-RPA32 (phospho Ser4 and phospho Ser8) antibody was purchased from Novus Biologicals LLC (Centennial, CO) and anti-phospho-histone H2A.X (Ser139) (20E3) antibody from Cell Signaling Technology, Inc. (Danvers, MA). TTK inhibitor "CFI402257" was purchased from MedChemExpress LLC (Monmouth Junction, NJ). For evaluation of *TTK*, *TPX2*, and *RAD54* expression by real-time PCR, TaqMan probes from Applied Biosystems, Inc. (Beverly, MA) were used.

**Cell viability.** Cell viability was assessed using Cell Titer-Glo Luminescent Viability Assay kit (Promega Corporation, Madison, WI).

**Evaluating impact on spontaneous DNA breaks and DNA end resection.** Impact of transgenic modulations on DNA integrity was monitored by evaluating cells for γ-H2AX, a marker for DNA breaks, using western blotting. Impact on DNA end resection, a critical step in the initiation of HR, was monitored by evaluating phosphorylation (on ser4 and ser8) of RPA32[20].

**HR activity.** HR activity was measured in a plasmid substrate, using a luminescence-based assay reported by us previously[14,17,42]. The HR substrate plasmid comprises two segments of a firefly luciferase gene, which are separated by an *Ampicillin* resistance gene. The segments of a firefly luciferase gene share a region of homology that serves as a substrate for HR, which generates a functional firefly luciferase gene by removing *Ampicillin* resistance gene. Gaussia luciferase gene in the plasmid, which is not affected by recombination, serves as an internal control. This plasmid is transfected into cells and, following incubation of cells for an appropriate duration, the cells are collected and HR assessed from the ratio of firefly and gaussia luciferase activities.

**Micronucleus assay.** Micronuclei (marker for genomic instability[19]) were evaluated by flow cytometry, using a commercial kit as reported by us previously[42].

**WGS and SNP arrays.** The control and transgenically modulated cells were cultured for different durations, genomic DNA purified, and analyzed using SNP6.0 arrays (Affymetrix), Axiom™ Precision Medicine Diversity Arrays, or WGS platforms. For each experiment, the parental or "day 0" cells (saved at the beginning of experiment) were used as the baseline genome to detect changes in control and transgenically modulated cells, during their growth in culture. SNP and WGS data were analyzed as described by us previously[14,16,22,23]. For evaluation of mutational signatures in each sample, the contributions of the mutational signatures were calculated using signal (https://signal.mutationalsignatures.com) with esophageal dataset[46] as the reference set. We have uploaded the data to Harvard Dataverse.

**Evaluating cell growth in a subcutaneous tumor model.** Six-week-old female Fox Chase SCID (CB17/Icr-Prkdcscid/IcrIcoCrl) mice were maintained as per guidelines of the Institutional Animal Care and Use Committee (IACUC); all experimental protocols were reviewed and approved by the IACUC and the Occupational Health and Safety Department of Dana Farber Cancer Institute, Boston, MA. The mice were exposed to 150 rads x-irradiation and injected subcutaneously with $5 \times 10^6$ EAC (FLO-1) cells, either untransduced (for evaluation of TTK inhibitor) or transduced with shRNAs (control or those targeting *TTK*, *TPX2*, and *RAD54B*). Transduced cells were selected in puromycin and cultured for another 2 weeks prior to inoculation in mice. For evaluation of TTK inhibitor, following appearance of palpable tumors, mice were treated with vehicle control or inhibitor (5 mg/kg injected intraperitoneally five times a week for 3 weeks). Tumor sizes were measured at least three times every week.

**Statistics and reproducibility.** For identification of differentially expressed genes, RNASeq raw counts of TCGA data were downloaded and pre-processed separately, using the package TCGAbiolinks. Each subtype expression matrix was normalized using the TCGAanalyze_Normalization function encompassing the EDASeq protocol. Differential expression was computed with the TCGAanalyze_DEA function,

implementing the EdgeR protocol. A cutoff of 1 was set for log2-fold change (in EAC relative to average of normal samples) with an FDR of 0.01, to identify genes overexpressed in EAC. Genomic instability in each patient was assessed from the total number of copy number events, which were computed using an R script. A copy event was defined as "high" if the segment mean was ≥2.5 and "low" if the segment mean was ≤1.5. Overall survival was defined as the day of last follow-up or day of death depending on vital status. Samples with a value of "NA" in the vital status row were excluded. Univariate analysis was performed using Cox regression with gene expression as the independent variable. For each gene, we recorded regression coefficient and hazard ratio with a *P*-value cutoff of 0.05. Experiments were conducted in replicates and figure legends provide information about the error bars and two-tailed *p*-values. For validation studies, both the gain and loss of function, two or more cell types, and the evaluation of multiple parameters of genome stability and growth (using different approaches) were used. The impact of overexpression of candidate genes in both the normal cells (which have low expression of these genes) and cancer cell lines (with already high expression of these genes), as well as the suppression of these genes in cancer cell lines was investigated. Multiple approaches and methods to evaluate different parameters of genome stability included the evaluation of spontaneous DNA breaks, DNA end resection, HR activity, micronuclei, and acquisition of genomic changes over time using SNP arrays and WGS to demonstrate that increased expression of *TTK*, *TPX2*, and *RAD54B* disrupts genomic integrity and stability.

**Reporting summary.** Further information on research design is available in the Nature Research Reporting Summary linked to this article.

## Data availability

Whole-genome sequencing data have been deposited on NCBI Submission Portal. The BioProject accession number is PRJNA706422. Single-nucleotide polymorphism data are deposited at publicly available website Harvard Dataverse with doi: 10.7910/DVN/STUEW0 and url: https://doi.org/10.7910/DVN/STUEW0. Source data underlying the graphs and charts presented in the main figures are available in Supplementary Data 1. Full blot/gel images are presented in Supplementary Data 2. Clinical information related to esophageal adenocarcinoma (ESCA) dataset GSE19417 is provided in Supplementary Data 3. Clinical information related to the ESCA TCGA dataset is available in Supplementary Data 4.

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

## Acknowledgements

This work was supported by Department of Veterans Affairs Merit Review Award I01BX001584-01 (N.C.M.), NIH grants P01-155258 and 5P50 CA100707 (M.A.S., M.K.S., N.C.M.), and Leukemia and Lymphoma Society translational research grant (N.C.M.).

## Author contributions

M.A.S. envisioned the study, analyzed and interpreted data, and prepared manuscript. N.C.M. assisted in data interpretation and critical review of manuscript. L.B., M.R., and M.K.S. conducted bioinformatic and statistical analyses. S.K., L.B., and S.T. equally contributed to major experiments and manuscript preparation. C.L., J.S., C.C., G.B.G., Y.T., and R.P. contributed to specific experiments and data analyses.

## Competing interests

The authors declare no competing interests.
