## [Peer Review File · Communications Biology]

Reviewers' comments:

Reviewer #1 (Remarks to the Author):

The authors utilised existing TCGA datasets to perform a customised integrative analysis that takes into account copy number variation, transcriptomic changes and survival, and found genomic instability gene signature to be overexpressed in the EACs samples. The observation of genomic instability is consistent with other publications (PMID: 28930282, PMID: 23604115) Of all the genes identified that supposedly contribute to genomic instability, the authors chose 3 for further in vitro and in vivo studies. The data established that these 3 genes accelerate tumour growth in EAC.

Of all the genes found to contribute to genomic instability, it will be informative if the authors can classify them according to the roles they play in genomic instability - which aspect of genomic instability they are enriched in EAC? In addition, the authors should justify why the 3 genes -TTK, TPX2, RAD54B are chosen for further studies.

The authors have selected 2 EAC cell lines for their downstream studies, but the expression of TTK, TPX2, RAD54B are known to be high in these cell lines to begin with, which may compromise the strength of subsequent functional genomic analysis. It will be more powerful if the authors could involve and screen more EAC cell lines (which are available), and then select cell lines with low and high expression of these 3 genes for their downstream studies. Another question is whether overexpression of these 3 genes in normal non-cancerous cells also increases genomic instability, or their function is dependent on the context of cancer. In vivo TTK-I treatment does reduce tumour volume, but again, addition of cell lines with low expression of the 3 genes will support evidence of the efficacy of the inhibitor.

It will be helpful to the readers if the authors can reorganise the data and figures to improve readability, flow and clarity

Reviewer #2 (Remarks to the Author):

Through integrative genomic data analyses and in-depth in vitro and in vivo functional validations, Kumar and colleagues have identified several driver genes (e.g. TTK, TPX2 and RAD54B) whose overexpression are highly associated with genomic instability in esophageal adenocarcinoma. Interestingly and importantly, they also showed that TTK inhibitor synergistically increases chemotherapy-induced cytotoxicity while inhibiting rather than promoting genomic instability in surviving cells. This is a well-designed study providing strong and clean evidences on the drivers of genomic instability in EAC. I have one major comment and several minor ones.

Major comments:

1. In the last section of Results, the authors performed experimental assays on HR activity and DNA breaks to show that TTK inhibitor inhibits spontaneous DNA damage and HR activity, and reverses genomic instability caused by chemotherapeutic agent. Although these experiments are essential and important, genomic sequencing (e.g. WGS/WXS) experiments are also requisite in order to verify these important findings. For instance, would the copy number/mutational burden be decreased in combined TTK inhibitor and chemo treatments as compared to chemo agent alone?

Minor comments:

1. Fig. 5C – II, the mutational signatures in normal control should also be shown.
2. It's unclear what the colors of bars in supplementary fig 2 represents. Also, why some bars have

two layers of asterisks with different colors? These should be described in the fig legend.

3. It's unclear to me why gene expression data in only 11 TCGA normal samples were used while there were 88 EAC tumor samples. Are the expression data in many matched normal samples not available?

4. I suggest supplementary fig 5 to be moved into main text and combined with Fig. 6. This is an important figure showing the synergism of TTK inhibitor and chemotherapeutic agents.

Point by point response to reviewers' comments and details of revisions.

Reviewer # 1 #1:

Comment 1: Of all the genes found to contribute to genomic instability, it will be informative if the authors can classify them according to the roles they play in genomic instability - which aspect of genomic instability they are enriched in EAC?

Response: We have now added a Table (as **New Supplementary Table 1**) providing information about roles of all 31 genes based on published as well as our own data (Mentioned on Page 3, 2nd paragraph, lines 5-6 in revised manuscript and shown as **Supplementary Table 1**).

Comment 2: should justify why the 3 genes -TTK, TPX2, RAD54B chosen for further studies.

Response: For practical reasons, in most screens, usually one or two hits are investigated in depth in a single paper. We chose three genes which belonged to diverse pathways of genome stability/growth. These included TTK, a kinase; TPX2, a spindle assembly factor; and RAD54B, a homologous recombination protein. **This information is now clearly provided in the paper (Revised manuscript: from last two lines of page 4 to first two lines of page 5).**

Gene Name	Family	Function/Pathway	Ref. #
ARHGAP11A	Rho GTPase	Cell cycle, DNA damage response	4
ARHGAP11B	Rho GTPase	Brain Development	5
BRI3BP	BRI3 binding protein	Involved in tumorigenesis	6
BUB1B	Serine/Threonine Kinase B	Mitotic checkpoint kinase	7
CAPRIN1	Cell Cycle Associated Protein	Cell proliferation	8
CASC5	Kinetochore protein	Cell cycle regulation	9
CCDC138	Coiled-coil domain-containing	Unknown	
CENB2	B-type cyclins	G2/M Cell cycle regulation	10
CCT6A	Chaperonin protein	Protein folding	11
CDK1	Cyclin dependent kinase	G2/M Cell cycle regulation	12
CENPO	Centromere protein	Regulation of mitosis	13
CSE1L	Nuclear export factor	Cell cycle and genomic instability	14
DKC1	Small nucleolar ribonucleoprotein	Cell cycle	15
ERCC6L	Mitotic helicase	Mitosis checkpoint regulation	16
FAM72B	Family with sequence similarity to kinetochore associated protein	Cell cycle regulation	17
KIF11	Kinetochore associated protein	Mitosis checkpoint regulation	18
KIF23	Kinetochore associated protein	Mitosis checkpoint regulation	18
KIF4A	Kinetochore associated protein	Mitosis checkpoint regulation	18
LEO1	RNA polymerase II associated factor Paf1	Oncogene	19
MST4	Serine/threonine protein kinase	Promote cell growth and transformation	20
NEK2	Mitotic kinase	Cancer progression	21
NUSAP1	Nucleolar & spindle associated	Promote cancer progression	22
PSMD14	Deubiquitinating enzyme	Promote tumor metastasis	23
RAD54B	DNA repair and recombination protein	Promotes homologous recombination	24
SMC2	Structural maintenance of chromosomes protein	Involved in DNA repair pathway and genomic instability	25
STIL	Centriolar replication factor	Involved in DNA damage response	26
STIP1	Stress induced phosphoprotein	Tumor growth, metastasis	27
YOMM34	Mitochondrial import receptor	Promotes cancer growth	28
TPX2	Microtubule-associated protein	Genomic instability, cancer	29
TROAP	Cytoskeleton, spindle assembly	Cancer and metastasis	30
TTK	Mitotic kinase	Homologous recombination and cancer growth	31

Supplementary Table 1. Known functions and pathways of GIS31 genes.

Comments 3 and 4: The authors have selected 2 EAC cell lines for their downstream studies, but the expression of TTK, TPX2, RAD54B are known to be high in these cell lines to begin

with, which may compromise the strength of subsequent functional genomic analysis. It will be more powerful if the authors could involve and screen more EAC cell lines (which are available), and then select cell lines with low and high expression of these 3 genes for their downstream studies. Another question is whether overexpression of these 3 genes in normal non-cancerous cells also increases genomic instability, or their function is dependent on the context of cancer. In vivo TTK-I treatment does reduce tumour volume, but again, addition of cell lines with low expression of the 3 genes will support evidence of the efficacy of the inhibitor.

Supplementary Figure 4. Overexpression of TTK, TPX2 and RAD54B increases DNA breaks and HR activity in EAC (OE19) cells. A) OE19 cells were transfected with control plasmid (C) or those overexpressing TTK (TTK-O), TPX2 (TPX2-O) or RAD54B (RAD54B-O), selected in puromycin and evaluated for γ -H2AX and phosphorylated-RPA32, using Western blotting (I), and HR activity, using a plasmid-based assay (II). Error bars indicate SDs of experiments conducted in triplicate; Two-tailed p values: = p < 0.5; **B)** The transgene overexpression confirmed by Q-PCR (shown in Supplementary data).

Response: We initially used one normal esophageal cell type to study the impact of overexpression of these genes, one EAC cell type to study the impact of further increase (overexpression) of these genes and two EAC cell lines to study the impact of knockdown of these

genes on genome stability. **The genome stability data using SNP and whole genome**

Supplementary Figure 5A

Supplementary Figure 5. Overexpression of TTK, TPX2 and RAD54B increases DNA breaks and DNA end resection in EAC (FLO-1) cells. A) FLO-1 cells were transfected with control plasmid (C) or those overexpressing TTK (TTK-O), TPX2 (TPX2-O) or RAD54B (RAD54B-O), selected in puromycin and evaluated for γ -H2AX and phosphorylated-RPA32, using Western blotting; B) The transgene overexpression confirmed by Q-PCR (shown in Supplementary Material).

sequencing platforms, shown in Figures 5A-C and Supplementary Figure 3 is from

overexpression of these genes in

epithelial cells. We have now also shown the impact of the overexpression of these genes in EAC cell line (OE19) on spontaneous DNA breaks, DNA

end resection, and HR activity (New Supplementary Figures 4 A and B) as well as genomic instability (New Supplementary Figure 6). The impact of the overexpression on spontaneous DNA breaks and DNA end resection, a distinct step in the

Supplementary Figure 6. Overexpression of TTK, TPX2 and RAD54B increases genomic instability in EAC (OE19) cells. OE19 cells were transfected with control plasmid (C) or those overexpressing TTK (TTK-O), TPX2 (TPX2-O) or RAD54B (RAD54B-O), selected in puromycin and evaluated for micronuclei, a marker of genomic instability. Flow cytometry images of micronuclei (I) and bar graphs showing percentage of micronuclei (II) are shown.

initiation of HR, is now also shown in EAC (FLO-1) cells (New Supplementary Figure 5). We also suppressed these genes in normal cells (fibroblasts). However, since normal

cells have very low levels of expression of these genes and relevant activities, the knockdown did not produce any conclusive data, which was expected (not shown). So now, we have done overexpression of these genes in both the normal cells (which have low expression of these genes) and cancer cell lines (with already high expression of these genes) as well as suppression of these genes in cancer cell lines. Moreover, we have used multiple approaches and methods including the evaluation of spontaneous DNA breaks, DNA end resection, HR activity, micronuclei, single nucleotide polymorphism arrays and whole genome sequencing to demonstrate that increased expression of TTK, TPX2 and RAD54B disrupts genome stability (This information is provided in Lines, 5 – 16, Page 6 of revised manuscript and data shown in Supplementary Material).

Reviewer #2.

Major comments:

1. In the last section of Results, the authors performed experimental assays on HR activity and DNA breaks to show that TTK inhibitor inhibits spontaneous DNA damage and HR activity, and reverses genomic instability caused by chemotherapeutic agent. Although these experiments are essential and important, genomic sequencing (e.g. WGS/WXS) experiments are also requisite in order to verify these important findings. For instance, would the copy number/mutational burden

be decreased in combined TTK inhibitor and chemo treatments as compared to chemo agent alone?

Response: 1) We used both the WGS and SNP arrays to confirm impact of the overexpression of these genes in normal cells. For combination experiments, we demonstrated impact on genome stability by evaluating micronuclei (marker of ongoing genome stability), DNA breaks, and HR (a mechanism of ongoing copy number and LOH events in cancer; Shamma et al. 2009; Pal et al.). To further demonstrate the impact on copy number changes, we now demonstrate this using Axiom™ Precision Medicine Diversity Arrays. We show that etoposide increased the

Supplementary Figure 7. TTK inhibitor reduces etoposide-induced acquisition of copy number events in EAC cells. FLO-1 cells, control (C; DMSO only) or those treated with TTK inhibitor (TTKI-I; 10 nM) and etoposide (ET; 1 μM), alone as well as in combination with each other for 3 weeks. DNA from these and baseline control (day 0) cells was purified and acquisition of copy number events during growth of cells in culture vs. day 0 cells (representing baseline genome) monitored, using Axiom™ Precision Medicine Diversity Arrays; a copy event was defined as a change in ≥ 3 consecutive CNV probes by 1 copy. (I) Amplifications (red dots) and deletions (blue dots) on different chromosomes; (II) Bar graph showing copy number events throughout genome.

acquisition of copy number events relative to control cells, whereas addition of TTK inhibitor was

able to reduce/prevent this increase. (This is now mentioned in Lines 15-18, on Page 8 of revised manuscript but data is shown in Responses to Minor comments:

Minor Comment 1. Fig. 5C – II, the mutational signatures in normal control should also be

Figure 5C, II. Mutational instability and underlying processes. Mutations found in “day 0 sample” were removed from all samples; Representation of mutational processes extracted in all four samples. For each sample, the contribution of the mutational signatures were calculated using signal (<https://signal.mutationalsignatures.com>) with esophageal dataset⁴³ as the reference set. For each sample (color coded bars) number of mutations (y-axis) contributed by mutational signatures (x-axis) are shown.

Response: We have now re-processed the data using previously published esophageal cancer whole genome dataset as the reference, and now show mutational signatures also in control sample (presented in Lines 7 – 10, page 7; and New Figure 5C, II of revised manuscript).

Minor Comment 2. It's unclear what the colors of bars in supplementary fig 2 represents. Also, why some bars have two layers of asterisks with different colors? These should be described in the fig legend.

Response: Colors have been removed and asterisks have been defined in the figure legend (Please see revised Supplementary Figure 2 legend).

Supplementary Figure 2. Functional siRNA screen evaluating GIS31 genes for impact on homologous recombination (HR) activity in EAC cells. EAC (FLO1) cells were transfected with siRNAs, either control (non-targeting) or those targeting 31 potential genomic instability (GIS31) genes, and impact on HR assessed using strand exchange assay described in Methods section. Bar graphs show percent inhibition of HR activity; error bars represent SDs of three independent experiments. Two-tailed p-values, indicating significance of difference relative to control siRNA-transfected cells, are shown as: * < 0.05 — > 0.005; ** < 0.005 — > 0.0001; *** < 0.0001 — < 0.000005.

Minor Comment 3. It's unclear to me why gene expression data in only 11 TCGA normal samples were used while there were 88 EAC tumor samples. Are the expression data in many matched normal samples not available?

Response: Yes, TCGA only had 11 normal samples.

Comment 4. I suggest supplementary fig 5 to be moved into main text and combined with Fig. 6. This is an important figure showing the synergism of TTK inhibitor and chemotherapeutic agents. **Response:** This has been done.

REVIEWERS' COMMENTS:

Reviewer #1 (Remarks to the Author):

The revised paper addressed most of reviewres' concerns adequately and the manuscript is substantially improved. Great to see the authors for careful attention to all the reviewers' comments

Reviewer #2 (Remarks to the Author):

My concerns have been well addressed. I recommend it for publication.